# Benchmarking of an Intervention Aiming at the Micro-Elimination of Hepatitis C in Vulnerable Populations in Perpignan, France, to Inform Scale-Up and Elimination on the French Territory

**DOI:** 10.3390/v16101645

**Published:** 2024-10-21

**Authors:** Gordana Avramovic, Laura O’Doherty, Tina McHugh, Andre Jean Remy, Arnaud Happiette, Hakim Bouchkira, Philippe Murat, Olivier Scemama, Adrien Esclade, Maria Isabel Farfan Camacho, Walter Cullen, John S. Lambert

**Affiliations:** 1Mater Misericordiae University Hospital, 44 Eccles St, D07 R2WY Dublin, Ireland; gavramovic@mater.ie (G.A.);; 2Catherine McAuley Centre, University College Dublin, 21 Nelson St, D07 A8NN Dublin, Ireland; 3Centre Hospitalier de Perpignan-20 Avenue du Languedoc, 66046 Perpignam, Franceaccueil1.emh@ch-perpignan.fr (A.H.);; 4Ministère de la Santé et de la Prévention, 14, Avenue Duquesne, 75007 Paris, France; 5European Commission, Directorate-General for Structural Reform Support, CHARL 9/100, B-1049 Brussels, Belgium

**Keywords:** hepatitis C, vulnerable populations, people who inject drugs, PWID, integrated care, systems of care, HCV elimination, benchmark

## Abstract

Hepatitis C virus (HCV) is an important cause of chronic liver disease. Among at-risk populations, access to care is challenging. The French Ministry of Health has supported a seek-and-treat pilot intervention aiming at micro-elimination in Perpignan, France, to inform scale-up of elimination efforts across the whole territory. University College Dublin (UCD) led a successful EU funded project, called HepCare, focusing on the micro-elimination of HCV. UCD was contracted to evaluate and benchmark the Perpignan results against results from HepCare. Using mixed-method approaches including qualitative interviews with patients, a focus group with healthcare professionals, and quantitative analyses of the cascade of care against results obtained at other European sites, we analyse the acceptability, reproducibility, replicability, and effectiveness of the Perpignan intervention. A total of 960 participants were recruited in the Perpignan area. HCV antibody test results were obtained for 928 (96.6%), of which 150 (15.6%) were antibody-positive. Of the antibody-positive participants, 68 (45.3%) tested positive for HCV-RNA, 141 (94%) were linked to care, and of the HCV-RNA-positive participants, 60 (88%) started treatment. Of those who underwent treatment, 34 (56.7%) completed treatment and achieved a sustained viral response (SVR) at dataset closure, 18 (30%) were still in treatment, 5 (8.3%) defaulted from treatment, and 3 (5%) had a virologic failure or died. The intervention in Perpignan was acceptable to patients, but had limitations in effectiveness, as shown in comparisons with HepCare results. To engage harder-to-reach cohorts in France, future models of care in the territory should incorporate peer support.

## 1. Introduction

Hepatitis C virus (HCV) infection is one of the most common causes of chronic liver disease worldwide. HCV is primarily transmitted through injecting drug use (IDU) and medical interventions, such as blood transfusions and immunization [1]. In 2015, it was estimated that approximately 150–175 million people were infected with HCV [2]. In 2017, the World Health Organisation launched the “Global Health Sector Strategy (GHSS) on viral hepatitis 2016–2021”. This strategy aims to eliminate viral hepatitis as a global health threat by 2030 [3].

Vulnerable populations, including people who inject drugs (PWID), are at a high risk of developing chronic HCV infection [4]. Often these individuals do not seek specialised care in a traditional manner, and flexibility and portable treatment options are essential to meet their needs [5]. Despite now having highly effective direct-acting antiviral (DAA) oral treatments available to treat HCV infection, there continues to be a challenge in reaching individuals who may not yet show symptoms of disease. To better address this issue, and ultimately achieve the aim of eliminating chronic HCV by 2030, ‘Test to Treat’ systems and pathways within communities have been established to target high-risk individuals, diagnose them, and link them to care for treatment [6].

In 2016, University College Dublin/Mater Misericordiae University Hospital (UCD) led the HepCare Europe Project, a three-year project co-funded by the Third Health Programme of the European Union involving sites in Ireland, the United Kingdom, Spain and Romania [7]. The project was a test-to-treat intervention targeting PWID and those with a history of IDU, who were considered at risk of HCV infection and disease progression. UCD’s role in HepCare Europe was to coordinate the overall project and evaluate it as a system of care [8], using quantitative and qualitative data generated from its four European sites.

In 2018, the French Ministry of Health (MoH) initiated a test-to-treat pilot in Perpignan to inform a later larger-scale implementation of Test-to-Treat interventions in the French territory. Perpignan is situated in the South of France near the Spanish border and was chosen because it has a high proportion of people on low income (30% of the population lives on social welfare), a high prevalence of people who use drugs, and the cost of drugs there is low (5–10 euros for a dose of cocaine).

Based on expert experience gained through leading and coordinating HepCare Europe, UCD was selected by the Directorate-General for Structural Reform Support (DG REFORM) programme of the European Union to support the French MoH and evaluate the pilot.

The data and methods from HepCare are used to evaluate the French intervention in Perpignan and provide an international point of comparison to the MoH. The hypothesis tested is that the Perpignan intervention is a suitable model for the micro-elimination of HCV, which could be replicated in other French territories.

## 2. Materials and Methods

### 2.1. Setting

The intervention was conducted in Perpignan, in addiction services (Centres de Soins d’Accompagnement et de Prévention en Addictologie–CSAPA), harm reduction services (Centres d’Accueil et d’Accompagnement à la Réduction des risques pour Usagers de Drogues–CAARUD), restaurants, healthcare settings, emergency accommodation, and social centres.

### 2.2. Study Population

The target population were PWID from vulnerable groups. The approach from the Perpignan was to ‘reach out’ to PWIDs, first in drug centres, and then in the places where they live: hostels, squats, and food distribution centres. The treatment strategy was to go from diagnosis to treatment in one day.

### 2.3. Study Design

This study employed a pilot design: it is a feasibility study of a Hepatitis C screening intervention for vulnerable populations, in which screening was carried out in community settings via a mobile health unit (MHU). The MHU was staffed by an outreach nurse and mediator, and periodically by a social care worker. The MHU travelled to community sites that the target population were known to frequent, and patients were offered the intervention. Services offered on the MHU included antibody (Ab) tests, RNA rapid tests via a CEPHEID machine, and diagnostic tests such as ultrasound elastography, used to measure the extent of liver damage. A telemedicine link was used to involve the hospital specialist, who provided clinical oversight and determined the medication regime required. MHU staff provided continuity and support to patients throughout the treatment duration. This pilot took place between 2018 and 2022.

The methodology for the evaluation evolved based on a literature review conducted by UCD regarding current initiatives surrounding integrated hepatitis C models of care following advancements in treatment regimens and diagnostic technologies. UCD then developed indicators for the evaluation of the French pilot based on indicators collected for HepCare. The indicators produced were discussed and agreed with the French Ministry of Health (MoH) and the National Health Insurance Fund (CNAM), who financed the intervention. Three types of analyses were conducted, modeled on already published HepCare methods: qualitative [9] patient interviews to evaluate the acceptability of the intervention to the target population; quantitative data analysis of the cascade of care results [7]; and a focus group with healthcare professionals, which was designed to explore the strengths, limitations, replicability, and reproducibility of the intervention. The results were then benchmarked against results collected by the HepCare Europe study in four European cities to evaluate the effectiveness of the intervention.

### 2.4. Data Collection and Analysis


*Pilot data collection*


The Perpignan team collected and aggregated pilot data and provided anonymised databases of results to UCD for evaluation.


*Evaluation data collection*


Qualitative patient interviews: Semi-structured interviews took place via Skype due to the COVID-19 pandemic.

Inclusion criteria: Participants were required to be over the age of 18, capable of providing informed consent, and were required to have a positive HCV-RNA result. The HCV-RNA test was used to measure the viral load of hepatitis C in the blood to indicate whether a participant’s state warranted hepatitis C virus treatment. Exclusion criteria: Patients who tested positive for the HCV antibody result but negative for HCV-RNA were excluded from the interviews; however, their participation was included in the data collection and consent taken for other aspects of the study. Those patients were excluded because, although they had been exposed to HCV at some point in the past, they had either spontaneously resolved/cleared the infection and did not require treatment or they had been previously treated successfully elsewhere, and we focused on those treated within the pilot for the qualitative interviews, in order to gather information on the whole process.

Interviews were delivered by a French native speaker, audio-recorded, transcribed in French, and translated into English. Transcripts were then circulated to members of the research team for qualitative analysis. An inductive thematic analysis was conducted in order to identify emerging themes from the data. Two researchers coded the data individually and collaborated at a later stage to compare findings. This process helped to mitigate subjective reasoning and potential bias. Once guiding themes were agreed upon by the researchers, they categorised them into sub-themes.

Focus group with healthcare professionals: A focus group with healthcare professionals implementing the Perpignan intervention was held via an online platform (Zoom Pro) due to COVID-19 restrictions. All healthcare professionals from the Perpignan hospital who implemented the intervention were invited to participate in interviews. The facilitator shared the semi-structured interview questions with participants in advance to give them the opportunity to reflect on their experiences and provide considered input during the interview itself. Interviews were held by a native French speaker, audio-recorded, transcribed in French, and translated into English. Transcripts were distributed to members of the research team for analysis. Qualitative outcome analysis was used to amalgamate the findings from the data. The rationale for using this methodological approach was that it assisted the researchers to evaluate the outcome of the intervention and to later propose ideas for new and/or enhanced interventions. Furthermore, a brainstorming session was organised internally by the UCD team to look at possible ways to replicate the intervention in other French regions with a different prevalence using their expertise. All members of the UCD team who had participated in the HepCare project were invited to comment on the Perpignan intervention.

Quantitative data on the cascade of care: Data were collected by the Perpignan hospital site and anonymised aggregated data were released to UCD for the requested quantitative indicators. Gaps and inconsistencies in data were queried by UCD and addressed by site personnel. The quantitative data were compared with results obtained by the HepCare Europe project. Due to discrepancies in the numbers recruited, a statistical comparison was favoured for analysis.

### 2.5. Ethics

UCD circulated the list of the quantitative and qualitative indicators to the Perpignan site and French authorities. The Perpignan site obtained ethical approvals to conduct the intervention and the intervention evaluation.

## 3. Results

### 3.1. Qualitative Patient Interviews (See Appendix A)

Thirteen semi-structured interviews were administered to participants. A total of nine male and four female participants participated in semi-structured interviews. Three main themes emerged from the qualitative interview: access to new DAA treatment enabled engagement with treatment and treatment completion; intervention staff encouraged patient engagement and retention with Test and Treat programme; and receiving HCV treatment led to additional positive outcomes for patients. The main themes showed that the intervention was acceptable to patients.

### 3.2. Quantitative Results on the Cascade of Care

#### 3.2.1. Community Outreach Sites (See Table 1)

The Perpignan initiative reached out to 44 community sites via the mobile health unit dispatched by the hospital as seen in Table 1. The sites were grouped by category to facilitate an international comparison. Here, French services that can be used by the homeless population were categorised together, although they may be accessible to other populations as well. The categorisation of homeless services was therefore high (71%) because of the inclusion of restaurants, food banks, and social town hall services. These are not clinical services and would include populations living below the poverty threshold, but not necessarily people who use drugs. Drug addiction centres represented 11% of the services reached, which is relatively low.

**Table 1 viruses-16-01645-t001:** Types of service that were outreached to by site.

	Bucharest Romania	Dublin/Cork	Seville	London	Total HEPCARE	PERPIGNAN
		Ireland	Spain	UK (Mobile Health Unit- MHU)		(MHU)
Homeless Services	0 (0%)	2 (11%)	0 (0%)	105 (61%)	107 (49%)	31 (71%)
Drug Addiction Centre	3 (27%)	1 (6%)	8 (50%)	64 (37%)	76 (35%)	5 (11%)
General Practice	0 (0%)	14 (78%)	3 (19%)	2 (1%)	19 (9%)	0
Prison	2 (18%)	1 (6%)	0 (0%)	0 (0%)	3 (1%)	0
Night Shelter	3 (27%)	0 (0%)	0 (0%)	0 (0%)	3 (1%)	0
Needle Exchanges	0 (0%)	0 (0%)	0 (0%)	0 (0%)	0 (0%)	0
NGO (Non Governmental Organisation)	2 (18%)	0 (0%)	5 (31%)	0 (0%)	7 (3%)	4 (9%)
Other (healthcare facilities, home visits, ED ^†^, Lost To Follow Up case finding project (VIRAEMIC), mental health, youth)	1 (9%)	0 (0%)	0 (0%)	2 (1%)	3 (1%)	4 (9%)
Total (% of all organisations.)	11 (100%)	18 (100%)	16 (100%)	173 (100%)	218 (100%)	44 (100%)

^†^ ED emergency department.

#### 3.2.2. Cascade of Care (See Table 2)

The cascade of care demonstrated a high uptake, with 960 participants screened in Perpignan. While this was a higher number than at any of the individual HepCare Europe sites, the rate of antibody-positive patients in the tested cohort was 15.6% (n = 150), much lower than the European sites used for comparison, which yielded an average proportion of 41.8% Ab positives in the populations included.

The number of patients needing treatment, as defined by a positive RNA in Perpignan, was 68 (45.3% of those who were antibody positive). This number was much lower than what was found at other European sites, which had an average proportion of 60.5% in their population. For those RNA positive in Perpignan (n = 68), 14 (21%) had an F3 or F4 score, showing severe liver scarring or cirrhosis. They will need closer monitoring, as advanced cirrhosis is life-threatening, and they may also require further assessments for other complications like liver cancer and be considered for a liver transplant.

The number of participants linked to care was very high, with 141 patients (94%) who were antibody positive (n = 150) linked. Linkage to care happened at different stages of the cascade of care in different countries due to the setup of the localised healthcare and community services. Within the context of the Perpignan pilot, linkage to care meant that patients were offered interventions, usually provided at the hospital. Here, all Ab-positive patients were offered follow-up interventions (Fibroscan test and RNA tests). Of the 68 patients who were RNA positive, 60 patients (88%) were put on treatment. Linkage to care and start-up of treatment were proportionately higher in the Perpignan populations needing treatment than was achieved in any of the HepCare Europe sites. Out of those who completed treatment and obtained sustained viral response (SVR) results, there were no virologic failures, and all 34 patients obtained SVR (100%), demonstrating the efficiency of DAA treatment when completed. Patients who were not treated were almost exclusively patients with no social rights.

**Table 2 viruses-16-01645-t002:** Cascade of care results by country versus Perpignan results.

	Romania	Ireland	Spain	England	Total HepCare	Perpignan
A = Individuals recruited	525	812	636	635	2608	960
B = Participants antibody test results recorded (% of A)	525 (100%)	772 (95.9%)	636 (100%)	635 (100%)	2568 (98.5%)	928 (96.6%)
C = HCV Ab positive results (% of B)	230 (43.8%)	257 (33.0%)	197 (31.0%)	390 (61.4%)	1074 (41.8%)	150 (15.6%)
D = RNA positive HCV infections ^†^ (% of C)	71 (30.9%)	162 (63.0%)	108 (54.8%)	346 (88.7%)	687 (60.5%)	68 (45.3%)
E = Participants linked to care (% of C)	151 (65.6%)	176 (68.5%)	104 (52.8%)	219 (56.1%)	650 (64%)	141 (94%)
F = Participants put on treatment (% of D)	24 (33.8%)	104 (64.2%)	76 (70.4%)	115 (33.2%)	319 (43.5%)	60 (88%) ^†^
G = Still on treatment/RVS results pending (% of D)	4 (16.7%)	44 (42.3%)	20 (26.3%)	40 (34.8%)	108 (33.9%)	18 (30%)
H = Completed treatment SVR obtained (including virologic failure and death) (% of D)	20 (83.3%)	58 (55.7%)	54 (71.0%)	71 (61.7%)	203 (67.9%)	34 (56.7%)
I = Abandoned treatment (% of D)	0 (0%)	2 (2%)	2 (2.6%)	4 (3.5%)	8 (2.7%)	5 (8.3%)
J = Achieved SVR vs. put on treatment (% of F)	18 (75%)	57 (54.8%)	52 (68.4%)	69 (60%)	196 (61.4%)	34 (56.7%)
K = Achieved SVR vs. completed treatment (% of H)	18 (90%)	57 (98.3%)	52 (96.3%)	69 (97.2%)	196 (96.5%)	34 (100%)
L = Virologic failures vs. completed treatment (% of H)	2 (10%)	1 (1.7%)	2 (3.7%)	0 (0%)	5 (2.5%)	0 (%)
M = Death during treatment/before SVR (% of F)	0	0	0	2 (2.8%)	2 (0.1%)	3 (5%)

^†^ The OST prescribing GP practices in Ireland referred all antibody positive patients to the hospital for RNA testing. Romania sent all antibody positive patients to the hospital for RNA testing. The RNA testing was performed in the MHU in France, so this was not conducted at the same stage of the cascade of care at all sites.

#### 3.2.3. Enrolled Population Characteristics vs. RNA Positive Population Characteristics

The patient population recruited was predominantly male, representing 72.1% of the total cohort (692). In the RNA-positive population, 50 patients were male (73.5%) and 18 were female (26.5%). The high proportion of male participants in Perpignan is in keeping with results from HepCare, where there also existed a majority of male patients, with 335 (84%) among those being RNA-positive.

The risk factors displayed in Table 3 compare risk factors in the targeted population with those who were RNA-positive in Perpignan. The risk factors in the RNA-positive patients were proportionately different compared to the population targeted for inclusion in the study. For those needing treatment, the main risk factor was IDU (76.5%), and secondly nasal drug use (13.2%). Other risk factors only accounted for 10.3% of the RNA positives combined. However, the profile of the overall population targeted only had 184 (19.8%) IDU.

The overall number of RNA-positive patients found in the targeted population was the lowest compared to HepCare sites, with only 7.1% found in the population included in the intervention (see Figure 1). For the RNA-positive cases (n = 68), 42 (62%) were already known cases, and 26 (38%) were new cases. Figure 2 shows the proportion of new cases found at HepCare sites. Depending on the proportion of new cases, each site may benefit from focusing more on case finding if the majority are new cases, or on re-engagement and support through care if most cases are known.

Intervention limitations were considered as follows. COVID-19 disruption limited the study implementation, requiring increased time to inform and educate patients; however, the intervention continued while hospital HCV services remained unavailable. Concerns were expressed with regard to the targeting of appropriate outreach sites for the intervention. There was also a requirement for ongoing POCT (point-of-care testing) training at outreach sites. In France, only medical biologists, doctors, nurses, midwifes, and laboratory technicians are legally permitted to perform testing. Other healthcare professionals (HCPs) need to receive formal training in order to conduct testing. The use of POCT by community peers can only be carried out within the framework of a derogatory experiment, in particular within the framework of Article 51, and an agreement with a medical biology laboratory that guarantees the quality of the examinations performed. This involves an administrative process for sites, which is considered a barrier in terms of staff time and priority. In practice, despite a considerable time investment from the MHU staff in training outreach staff to conduct POCT, the burden of the POCT screening lay with the MHU team. This was in part due to the high turnover of staff in community settings. Additionally, the hospital in Perpignan has been actively involved in HCV outreach activity since 2013, and had subsequently dealt with a number of historical cases. This work has had a positive impact on seroprevalence in the region and thus limited the number of patients recruited for treatment as part of this intervention.

### 3.3. Healthcare Professional Focus Group (See Appendix A)

Model replicability was explored during a focus group with questions asked regarding transferability of tools, staff competencies, and services, bearing in mind the associated cost-effectiveness of potential changes to the intervention.

The possibility of a national dedicated MHU was explored, but the Perpignan team were of the view that this would not be logistically feasible due to the size of the French territories. However, a dedicated regional MHU was considered to have real potential in a future HCV model of care. The Perpignan team suggested that while there are no comparable MHU models in the territories, there are street units (vehicles kitted out with some medical equipment), available on request. However, such vehicles do not have specific staff available to operate them. In theory, it is possible to draw on this resource; however, this is not the sole mission and purpose of the street units, and as such, would require vigorous collaboration and agreement to garner involvement and stakeholder buy-in.

Staff replicability: The role of the nurse was reflected on, and the potential for enhanced use of this role was identified during interviews. As it stands, in each gastroenterology service in France, there are dedicated HCV nurses in situ. This nurse role has an outreach remit, exists across the whole territory, and is already funded by the state, and as such would not require additional resourcing. However, there is little to no emphasis placed on the outreach aspect of this nurse role, and although funding for outreach is provided by the state, it is usually absorbed elsewhere in hospital budgets and not used for the purpose of assertive engagement of HCV patients. In a proposed future model of HCV care, the nurses affiliated to gastroenterology services would need to be upskilled and equipped with the essential tools to deliver services in the community. The nurse would also require training and support to identify and establish partnerships with community organisations. Finally, the option of a peer support worker who has lived experience with HCV was flagged as a possibility by UCD during the interviews with Perpignan staff, as this was piloted on the HepCare project and is an intervention that has been proven successful in the international literature. The option to include a peer had only been explored as a voluntary unpaid position by the team in Perpignan and an appropriate candidate could not be recruited, so the idea was not pursued.

## 4. Discussion

### 4.1. Key Findings

The Test and Treat HCV programme made a positive contribution to meeting the needs of the target population in Perpignan. However, the overall population recruited had a low prevalence of HCV, limiting intervention efficiency, with only 7.1% included for treatment. There was also a significant discrepancy in the risk factors present in the population included for enrolment and in the populations needing treatment. All patients who had a sustained viral response (SVR) test carried out at the end of treatment were proven cured. The majority of the RNA-positive patients were previously known cases, demonstrating the importance of focusing on re-engagement. HCP interviews highlighted potential for reproducibility across the French territory through restructuring the role of hospital hepatology nurses to include a more concentrated focus on outreach activity, and the opportunities for using a regional MHU model for future micro-elimination efforts. In the delivery of HCV treatment programmes, there are additional cost-effectiveness benefits to sharing resources.

### 4.2. International Comparison and Comparison with Other Literature

With the current model, the full Perpignan mobile team travels to outreach sites, spending considerable time conducting Ab-tests, enrolling patients into the intervention, and performing training of both patients and community organisations. The populations chosen for inclusion were removed from the key risk factor for HCV, which is PWID. Moreover, the rate of patients who abandoned treatment was higher than for the HepCare sites, with 5 patients (8.3%) abandoning treatment compared with 2.7% for the overall HepCare intervention. UCD assessed that the intervention could be improved through the use of a trained peer, who would target key PWID populations through their lived experience and have a higher capacity to infiltrate networks by recruiting patients at street level. Peers could operate from lists of Ab-positive patients provided by community structures, track them down, provide education, dispel patient fears around treatment, call patients when they begin to default from treatment, and organise support. They are also able to operate on their own to find patients, increasing cost-effectiveness. It has been widely demonstrated that the peer role is key to engaging vulnerable populations in care [10,11,12].

In other HepCare Europe countries, peers were trained to perform the Ab-tests. Currently, this should be possible in France, as it is possible to undertake official training to administer this non-invasive test (i.e., rapid diagnostic test). In addition, in Perpignan, most of the Ab-positive cases were known. Out of 150 Ab-positive patients, 107 (71.3%) had a known serology and 43 (28.7%) were identified as new cases. Therefore, most cases could have been recruited from existing results, thus limiting the need to perform an Ab test. Peers have been shown not only to increase the number of people who initiate therapy, but also the proportion who complete treatment [13]. peers were part of the clinical team at other HepCare sites, and were hired into a professional position (i.e., they were not unpaid volunteers, as was thought of by Perpignan staff). This offered more stability in the position and a career pathway at an advanced stage of the recovery process. Moreover, HepCare has conducted a cost-effectiveness study, showing the cost-effectiveness of peer support [14,15]. The evaluation has supported a further exploration of this role and its introduction in France.

In Dublin, the limited healthcare budget and high cost of DAA regimens restricted the availability of treatment from July 2017 to February 2018, with a freeze on new treatments imposed by the government. This significantly disrupted the comparable Irish HepCare cascade of care among targeted vulnerable populations. There are no current limitations to treatment in France, and the Perpignan team was highly effective at initiating treatment for those who needed it. The success of DAA treatment was clearly seen, with 100% treatment success in those who have receivedan SVR result to date.

### 4.3. Best Practice Models of Care and Intervention Reproducibility

An intervention is deemed reproducible if the same results can be achieved with the same tools, which means obtaining consistent results using the same input data, computational steps, methods, and conditions of analysis [16,17]. In the case of the intervention in Perpignan, a key issue to reproduce the intervention is the prevalence of HCV within the French territories, meaning that the intervention would not find as many individuals in regions of lower prevalence compared to Perpignan. Currently, the available data in the literature suggest that there are still many undiagnosed individuals infected with HCV within the French territories [18]. In order to obtain similar results, robust seroprevalence data would be necessary and it is possible that different interventions should be considered depending on HCV prevalence in different regions. Interventions similar to Perpignan could be developed in areas of higher prevalence; alternatively, the possibility of having “visiting outreach teams” could be considered in areas of lower HCV prevalence. The Perpignan team rejected the idea of a national MHU as not being feasible for all regions of France. However, the HepCare team in Dublin had a visiting MHU from the UK for a period of 5 or 6 years, with visits for HCV testing lasting for up to 15 days. The MHU travelled to local homeless and prison services to organise screening sessions and assess vulnerable populations for HCV care. A similar option could be considered using a visiting MHU to support French territories with a lower prevalence, where maintaining a whole regional team is not cost-effective.

Among models that are considered to be best practice, the European Monitoring Centre for Drugs and Drug Addiction (EMCDDA) has highlighted a number of intervention projects, including HepCare and Perpignan [19]. The tools currently used by the intervention in Perpignan are widely recognised as best practice tools, including POCT [20], Fibroscan [21,22,23], and GeneXpert [24,25]. The cost-effectiveness analysis for Perpignan was conducted by Ernst and Young. Concerns were expressed regarding the intervention’s high cost. Therefore, UCD explored other possible models. Here, we suggest that where seroprevalence is high, a regional team controlled by the ARS could be evolved, consisting of an MHU, POCTs, Fibroscan, GeneXpert, a coordinator and/or mediator, and a peer support worker. The team could be centralised by the region and made available to support hepatology nurses. Such a regional initiative has been implemented in Ireland (ICORN), with a coordinator supporting several hepatology outreach nurses. A regional model has also been chosen in the UK, where the HepCare team, originally supporting one district of London, is now commissioned to cover the entirety of London, with their own new blood-borne virus (BBV) van, utilising an expanded team of staff members. The MHU includes testing for HCV but also HIV and HBV, and additionally covers asylum populations as well as traditional inclusion health populations.

We suggest that the role of the hospital gastro-hepatology nurse, currently funded to do outreach, but where no outreach is implemented, should be reviewed to conduct outreach or outreach funds reallocated. This role, already funded by the French government, would need to be reformed to include POCT training, and a partnership developed between doctors and nurses to allow nurses to conduct Fibroscan and GeneXpert assessments via a “delegation” from the doctor. This is now a possibility following the recent reform of nurse roles in France, which is undergoing a validation process at the national level. It is important to note that such a delegation has been feasible and implemented in Perpignan already. The nurse would need to also establish a link via telemedicine to the hospital specialist for cirrhotic patients and complex patients. It is possible to envisage a secondary link with a GP for non-cirrhotic patients who have not had previous DAA treatment and have no co-morbidities. In France, GPs are currently allowed to prescribe two DAAs under recommendations for the simplified management of chronic HCV from the High Health Authority (Haute Autorité de Santé), as published on 20 May 2019. This measure is part of the Plan priorité prévention 2018–2022, which aims to eradicate HCV within the French territories by 2025. Nurse implementation has been successful in Perpignan and also for the HepCare Europe project, and this has proven to be a cost-effective intervention [8,15,26,27].

Regarding the use of MHUs within the different territories, another possibility is to use existing street units, centralising their coordination from the regional outreach team and going out to rural areas if needed. All the necessary tools (POCT, Fibroscan, and GeneXpert) are portable and could be shared and added to existing units when they are available, and returned to a “central hub” after use. Existing street units have a mandate to look after vulnerable populations, and calling on this mandate could be enforced to ensure cooperation. Street units would mainly be used for transport purposes, but other medical assessments could be carried out if they were equipped for, for example, TB and HIV. The addition of other assessments has been shown to be desirable for vulnerable populations, including migrants. Such a model of a “one-stop shop” has been previously developed, whereby the treatment of addiction, HCV, and other medical conditions are fully integrated, including a peer worker [28,29].

### 4.4. Recommendations to the French Ministry of Health

1. Evolve peer support interventions: As prevalence is diminishing in the territory, remaining at-risk patients are harder and harder to reach. This is likely to be more cost-effective in the later stages of eradication than the use of full hospital MHU teams.

2. Review the role of the hospital hepatology nurse funded for outreach, and where no outreach is implemented, either enforce implementation or consider re-allocation of funds to a regional specialised hub.

3. Consider the establishment of regional elimination teams for regions of higher prevalence, and how those teams could support lower prevalence regions for micro-elimination over periods of time.

4. Consider the use of existing street units for HCV elimination.

## 5. Conclusions

To improve the overall effectiveness regarding the recruitment and reproducibility of the intervention, UCD provided a number of recommendations to the French Ministry of Health. One such suggestion was the introduction of peer support to improve the uptake of the intervention, along with measures to restructure staffing and resources. Peer support workers, through lived experience, are well placed to access and support the most vulnerable and marginalised populations and have been shown to be a very valuable part of a team needed to combat HCV infection. France is currently on track to achieve HCV elimination of HCV by 2030 [30]. The populations treated to date could be considered ‘low hanging fruit’, in that they tend to be the simplest to engage and treat. Populations that are more difficult to reach will continue to be inaccessible unless interventions like the one in Perpignan are supported further within the territories. Recommendations and data provided can be used for other international comparisons.

## Figures and Tables

**Figure 1 viruses-16-01645-f001:**
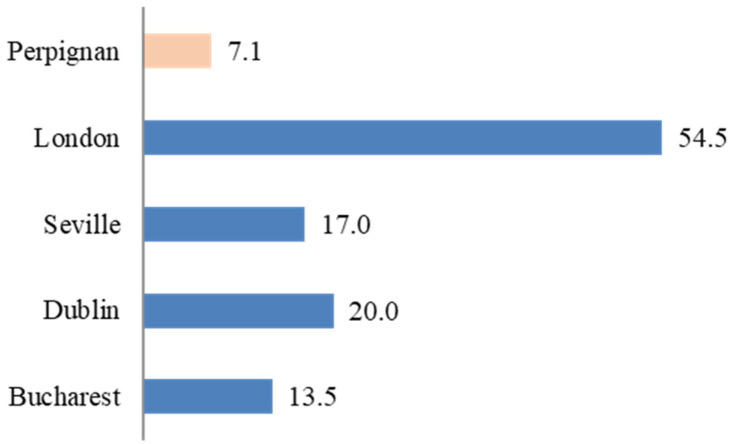
Percentage of RNA-positive patients in the targeted population.

**Figure 2 viruses-16-01645-f002:**
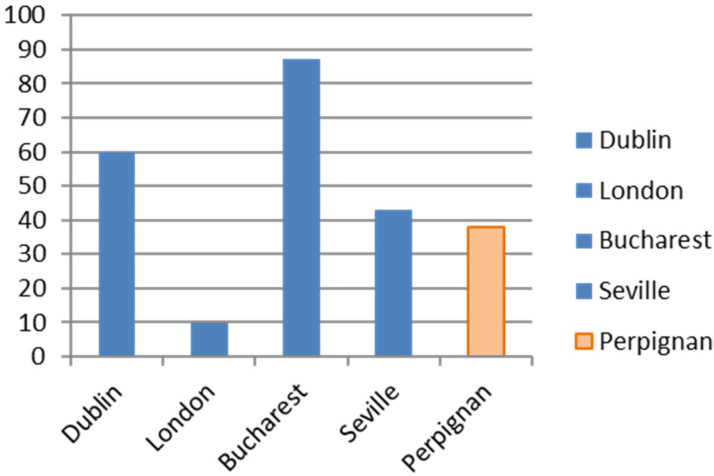
Percentage of new cases versus old cases.

**Table 3 viruses-16-01645-t003:** Risk factors in Perpignan: overall targeted population vs. RNA-positive patients.

Main Risk Recorded	Numbers Included in Study	% Included in Study	Number of Patients RNA Positive	% RNA Positive
Blood exposure (transfusion etc.)	68	7.3	2	2.9
Alcohol	116	12.5	4	5.9
CMU precariousness ^†^	151	16.3	0	0.0
Drugs, nasal	272	29.3	9	13.2
IDU	184	19.8	52	76.5
Prison > 1 year	26	2.8	1	1.5
Sexually transmitted infection (STI): infected partner	20	2.2	0	0.0
Geographic origin	46	5.0	0	0.0
Piercing and tattoos	26	2.8	0	0.0
Unprotected sex	19	2.0	0	0.0
Total	928	100%	68	100%

^†^ CMU (Couverture Medical Universelle) Universal Social Insurance for precariousness.

## Data Availability

Data available on request due to restrictions (legal).

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
