# Peer review of "Benchmarking of an Intervention Aiming at the Micro-Elimination of Hepatitis C in Vulnerable Populations in Perpignan, France, to Inform Scale-Up and Elimination on the French Territory"

_viruses, 2024, doi:10.3390/v16101645_

Round 1

Reviewer 1 Report

Comments and Suggestions for Authors

HCV elimination of HCV by 2030.”. 

Table S2: I don’t understand how the %s are calculated here, what is the denominator? Shouldn’t the denominator be the total within each site (sum of column?)

About other comments, please see the attachment.

Comments on the Quality of English Language

I appreciate that English is not the first language of the authors of this study, but this would benefit from a review by a native English speaker.

Author Response

Hello,

We have drafted our replies and we are presenting them in the attached word document. Thank you for your review,

Kind Regards,

Gordana Avramovic

Reviewer 2 Report

Comments and Suggestions for Authors

Please find below the comments and suggestions related to the manuscript viruses- 3174145.

Abstract section:

A total of 2608 participants were recruited across 218 clinical sites in Europe.

Some of the percentages below are not clear (how they were derived), such as antibody positive (41.8%), 687(60.5%), linked to care and 319(43.5%), etc.  

Also, it will be important to give an account of the 960 subjects recruited in the Perpignan study (and summarize the data obtained) since that data is also presented in the manuscript.

“HCV antibody test results were obtained for 2568(98.5%), 1074(41.8%) were antibody-positive, 687(60.5%) tested positive for HCV-RNA, 650(60.5%) were linked to care and 319(43.5%) started treatment. 196(61.4%) of treatment initiates achieved a Sustained Viral Response (SVR) at dataset closure, 108(33.9%) were still on treatment, 8(2.7%) defaulted from treatment, and 7(2.6%) had a virologic failure or died”.

All numbers and percentages should be determined from the original starting number (2568), or specified at each derived percentage (by mentioning ‘out of which’).  

Introduction section:

Reference numbering: It is not clear why the reference numbers are in Roman numerals. The standard numbering pattern for the Journal should be followed. Also, it is noticed that at some places, there is a mix normal and Roman numbers (when citing multiple references). Please edit accordingly.  

Please describe in brief what were the diagnostic and treatment methods employed in test-to-treat system in the Hepcare Europe project.

‘This data is used to analyse the French intervention in Perpignan’

It would perhaps be better to state as ‘The Hepcare Europe method was used to model the intervention strategy in Perpignan’

Materials and methods section:

Please provide information for Perpignan in brief (population size, location etc).

In the study design section, while the diagnostic tests are mentioned, the treatment strategy is not mentioned.

Please rephrase or edit the below statement for better clarity “…….. prior to the data collection stage giving the participants the opportunity to reflect on their experiences in advance of the interview to provide considered input during the process.”

What is the intervention referred to in the following statement?

“Furthermore, a brainstorming session was organised internally by the UCD team to look at possible ways to replicate the intervention using their expertise”.

Exclusion criteria: Patients who receive a positive HCV antibody result but not a positive HCV RNA

Please edit: Patients who test positive for HCV antibody result but negative for HCV RNA.

What did the ‘Quantitative data’ comprise of?

Exclusion criteria mentions ‘patients who receive a positive HCV antibody result but not a positive HCV RNA’ If an RNA positive result for HCV is the inclusion criteria, then would subjects with negative antibody result also be in the exclusion criteria?

Quantitative data on the cascade of care: Was the treatment provided at the sites where the initial data was collected through Mobile Health Units, or the subjects were asked to report to the Perpignan hospital?

Evaluation section:

Was the evaluation strategy not similar to the one used in the Hepcare Europe project? As  in the manuscript it is mentioned as ‘..The methodology for the evaluation evolved based on a literature review

conducted by UCD’

Ethics section

This section should also describe all the ethical approvals required and obtained before the commencement of the study, and the authorising body that provided the ethics approvals.

Results section:

Qualitative Patient Interviews:

Thirteen semi-structured interviews were administered to participants. Since the total subjects participated in the study was 960, does number of subjects interviewed (13) sufficient to represent the entire cohort of the study population?

Enrolled Population Characteristics vs RNA Positive Population Characteristics

‘Risk factors displayed in table S3 compare risk factors in the targeted population and

in the RNA-positive population in Perpignan’.

It should be risk factors in the targeted population who were tested RNA positive in the Perpignan population.

In table S3 – Please correct the spelling RNA ‘positive’.

In the RNA positive population (73.5% male and 26.5% female) can also be included in the Table S3.

The following statement is not clear, please elaborate.

‘The risk factors in the RNA-positive patients are proportionately different compared to the population targeted for inclusion in the study’.

The idea of a ‘peer support worker’ was not found to be feasible and the idea was not pursued. Yet, the Abstract section supports the idea of a peer support? The two statements do not appear to match.

Table 1:

How were the percentages (last column) for Perpignan population determined, and what do these represent?

Many of the percentages shown in each category do not represent the subject number in that category. For example, RNA positive subjects are 68 out of 960, which should be 7.1%, but is mentioned as 45.3%. This type of error appears in some other categories of subjects  and these discrepancies should be corrected. 

Discussion section:

A low prevalence of HCV should not be a determinant of ‘intervention efficiency’.

(as mentioned ……….a low prevalence of HCV limiting intervention efficiency with only 7.1% included for treatment). The intervention efficiency should be determined by the percentage of HCV positive subjects who received the treatment and were successfully cured of the infection.

End. 

Comments on the Quality of English Language

Minor editing required as mentioned in the comments.

Author Response

Hello,

We have replied in the attached word document. Thank you for your review,

Kind Regards,

Gordana Avramovic

Round 2

Reviewer 1 Report

Comments and Suggestions for Authors

The authors have addressed most of my previous concerns. However the following still require clarification:

1) The section regarding the "Evaluation data collection" is still unclear. Paragraph 1 describes qualitative patient interviews but does not discuss inclusion criteria for these interviews. Paragraph 2 describes focus groups with heatlth care professionals. Paragraph 3 & 4 describes inclusion/exclusion criteria but it is not clear what analysis these patients were included in - qualitative or quantitative? There is mention of qualitative interviews in this section, if so this should come alongside the patient interview paragraph (1). 

2) The cascade of care definitions (numerator/denominator for metrics) are still not described.

3) An English edit is required- for example:  Abstract: " of which anal150(15.6%) were antibody-positive".  study population: "The ttreatment strategy", etc.

4) Not all acronyms are defined in the tables

5) Recommend the use of best practices in terms of avoiding stigmatized language such as "harder-to-reach" 

Comments on the Quality of English Language

See above

Author Response

Here are our replies to the latest comments (round 2)  

1) The section regarding the "Evaluation data collection" is still unclear. Paragraph 1 describes qualitative patient interviews but does not discuss inclusion criteria for these interviews. Paragraph 2 describes focus groups with heatlth care professionals. Paragraph 3 & 4 describes inclusion/exclusion criteria but it is not clear what analysis these patients were included in - qualitative or quantitative? There is mention of qualitative interviews in this section, if so this should come alongside the patient interview paragraph (1). 

We have amended the section to clarify the inclusion/exclusion criteria by section.    

2) The cascade of care definitions (numerator/denominator for metrics) are still not described.

This is described-Letters were added to define the denominators-please see A, B, C etc and percentage of A, B, C, D, E etc. This clarifies the denominators.  

3) An English edit is required- for example:  Abstract: " of which anal150(15.6%) were antibody-positive".  study population: "The ttreatment strategy", etc.

Thank you. This was amended and the document was checked again.  

4) Not all acronyms are defined in the tables

Thank you. Those were defined when missing.  

5) Recommend the use of best practices in terms of avoiding stigmatized language such as "harder-to-reach" 

The position of the PI and other authors remains the same in the context of HCV. Hard-to-reach has been widely used and published.